# Variation in maternal mortality in Sidama National Regional State, southern Ethiopia: A population based cross sectional household survey

**Aschenaki Zerihun Kea**[1,2]*, **Bernt Lindtjorn**[1,2], **Achamyelesh Gebretsadik**[1], **Sven Gudmund Hinderaker**[2]

1 School of Public Health, College of Medicine and Health Sciences, Hawassa University, Hawassa, Ethiopia, 2 Centre for International Health, University of Bergen, Bergen, Norway

* aschenakizer@yahoo.com

## Abstract

### Introduction

Maternal mortality studies conducted at national level do not provide information needed for planning and monitoring health programs at lower administrative levels. The aim of this study was to measure maternal mortality, identify risk factors and district level variations in Sidama National Regional State, southern Ethiopia.

### Methods

A cross sectional population-based survey was carried in households where women reported pregnancy and birth outcomes in the past five years. The study was conducted in the Sidama National Regional State, southern Ethiopia, from July 2019 to May 2020. Multi-stage cluster sampling technique was employed. The outcome variable of the study was maternal mortality. Complex sample logistic regression analysis was applied to assess variables independently associated with maternal mortality.

### Results

We registered 10602 live births (LB) and 48 maternal deaths yielding the overall maternal mortality ratio (MMR) of 419; 95% CI: 260–577 per 100,000 LB. Aroresa district had the highest MMR with 1142 (95% CI: 693–1591) per 100,000 LB. Leading causes of death were haemorrhage 21 (41%) and eclampsia 10 (27%). Thirty (59%) mothers died during labour or within 24 hours after delivery, 25 (47%) died at home and 17 (38%) at health facility. Mothers who did not have formal education had higher risk of maternal death (AOR: 4.4; 95% CI: 1.7–11.0). The risk of maternal death was higher in districts with low midwife to population ratio (AOR: 2.9; 95% CI: 1.0–8.9).

### Conclusion

The high maternal mortality with district level variations in Sidama Region highlights the importance of improving obstetric care and employing targeted interventions in areas with

**Data Availability Statement:** The relevant data are available at Open Science framework: DOI https://doi.org/10.17605/OSF.IO/XVYR2.

**Funding:** This study was funded by Grand Challenges IDRC Canada through the Liverpool School of Tropical Medicine and REACH Ethiopia non-profit organization. The funders had no role in data acquisition, analysis, writing, and decision to submit for publication.

**Competing interests:** The authors have declared that no competing interests exist.

high mortality rates. Due attention should be given to improving access to female education. Additional midwives have to be trained and deployed to improve maternal health services and consequently save the life of mothers.

## Introduction

Ethiopia is the second most populous country in Africa having more than 110 million people [1] constituted by 11 regional states and two chartered administrative cities. The regional states and the two administrative cities are further divided into 800 woredas (districts). Important priorities on the government's agenda include Improving maternal health and consequently decreasing maternal mortality [2]. To improve maternal health and reduce maternal deaths, accurate data on maternal mortality should come from studies conducted at subnational level. However; the country's maternal mortality data mainly comes from studies carried out at national level [3, 4]. National maternal mortality estimates may not provide sufficient details to understand the distribution of maternal deaths at local levels relevant for health planning and monitoring. Hence, sub-national maternal mortality estimates are needed for program monitoring and local decision making.

To understand the distribution of maternal deaths at local level and improve the monitoring of the progress towards reducing maternal deaths, maternal deaths need to be accurately counted and the likely causes identified [5–7]. However, most developing countries do not have systems at national or sub-national level to register vital events including maternal deaths [8–10]. In areas without a vital registration system, maternal deaths can be measured through population based household surveys [11].

Most maternal deaths occur during labour, delivery or within 42 days postpartum. Important causes are obstetric haemorrhage, infections and hypertensive disorders of pregnancy [12]. Most of these deaths can be avoided through cost effective interventions including skilled birth attendance [13–15]. In countries with many maternal deaths, the coverage and usage of essential interventions is low, if available, often provided with poor quality, with a persisting gap between the rich versus the poor, and urban versus rural populations [16]. The Sustainable Development Goal (SDG) aims at reducing MMR to less than 70 per 100,000 live births (LB) by 2030, but this will not be achieved if universal coverage for essential interventions are not improved [5].

Since the launching of the Millennium Development Goal (MDG), the government of Ethiopia has taken measures to improve access to universal health coverage, emergency obstetric care and implemented other interventions focusing on maternal health services to reduce maternal mortality [17]. Hence, the maternal mortality ratio (MMR) was reduced from 1030 in 2000 to 401 in 2017 per 100,000 LB in the country [4]. Despite these improvements, still many mothers die [4]. The 2016 Ethiopia Demographic and Health Survey (DHS) reported a MMR 412 per 100, 000 LB [3]. Despite the prevailing problem, measuring maternal mortality remains a challenge in Ethiopia as the country lacks functional vital registration system [7]. Well organized household surveys, using large and representative samples with verbal autopsy (VA) can provide information on local distribution and causes of maternal deaths.

To the best of our knowledge, there have been few studies describing maternal mortality estimates and trends in reduction of MMR at sub-national and district level in the country. Population based studies conducted in south-west Ethiopia [18] and northern Ethiopia [19] found a MMR of 425 and 266 per 100,000 LB respectively. An implementation study from

south-west Ethiopia demonstrated a reduction of MMR by 64% during the intervention period from 477 to 219 deaths per 100,000 LB [15].

As there is no previous population-based study describing maternal mortality estimates and district-level variations in Sidama National Regional State, and as the principal investigator (AZK) is affiliated with Hawassa University, which is located in Sidama National Regional State, it was natural to conduct such a comprehensive study on this population. We carried out this study in the Sidama National Regional State, southern Ethiopia with the following specific objectives: 1) measure the maternal mortality ratio; 2) measure variations of maternal mortality ratio at district level; 3) assess determinants of maternal deaths.

This study could provide essential information to improve maternal health services relevant for lifesaving comprehensive emergency obstetric care in Sidama National Regional State. Furthermore, it will provide important information to the region used for priority setting and resource allocation identifying areas with high rates of maternal mortality. Therefore, this study can also inform other regional states in the country to carry out similar studies to understand the magnitude and variations in maternal mortality to improve maternal health care. The information from this study will help the design of maternal health programs at large which support the country's effort towards attaining the SDG.

## Methods and materials

### Study design and setting

We used a cross sectional study design employing population-based survey in households that reported pregnancy and birth outcomes in the past five years (July 2014-June 2019). The study was conducted in six woredas (districts): Aleta Chuko, Aleta Wondo, Aroresa, Daela, Hawassa Zuriya and Wondogenet of Sidama National Regional State, southern Ethiopia from July 2019 to May 2020. Sidama National Regional State is one of the 11 regional states in Ethiopia. The region had a population of 4.3 million people in 2020 [20] and administratively divided into 30 rural districts, 6 town administrations and 536 rural *kebeles* (the smallest administrative structure with average population of 5000). Under the *kebele*, there are local structures known as *limatbudin* (administrative unit organized by 40–50 neighbouring households).

The region has 18 hospitals (13 primary, 4 general and 1 tertiary), 137 health centres and 553 health posts operated by the government [21]. In the region, there are also 4 hospitals (1 general and 3 primary), 21 speciality and higher clinics, 131 medium clinics and 79 primary clinics run by private owners. The health centres provide basic emergency obstetric and new born care (BEmONC) whereas hospitals are responsible for comprehensive obstetric and new born care (CEmONC) in addition to the BEmONC [22].

### Study population and sampling technique

All women who experienced pregnancy and birth outcomes in the past five years in Sidama National Regional State were the source population. Women residing in sampled households and who had pregnancy and birth outcomes (live births, stillbirths and neonatal deaths) in the past five years preceding the survey were the study population.

Fig 1 shows the sampling strategy of the study. We followed multistage cluster sampling technique to select the study population. Probability sampling technique: the gold standard technique recommended to observe reliable findings (precision) was employed at each sampling stage [23]. In first stage, we listed all the 30 rural districts of the region with unique identification code. Then, we selected 6 districts (20% of the districts) by simple random sampling. At the second stage, we listed all the *kebeles* in the 6 districts and randomly selected 40 *kebeles* proportional to the size of the *kebeles* in the districts. We employed complex sampling

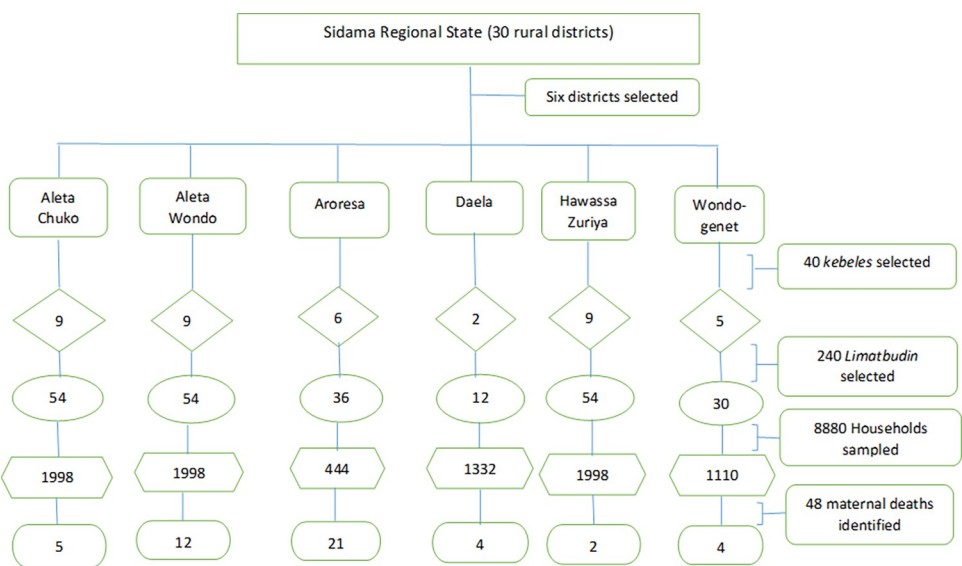

**Fig 1. Schematic diagram sampling techniques, Sidama Regional State, Ethiopia, 2020.**

technique and used seed number (245987) in statistical package for social science (SPSS) to generate the sample of *kebeles*.

In third stage, we listed all the *limatbudins* for each of the selected *kebele* and randomly selected 6 *limatbudins* from each *kebele;* altogether 240 *limatbudins* from the 40 *kebeles*. To identify a mother who experienced pregnancy and pregnancy outcomes in the past five years, we visited all the households in the selected *limatbudins* and listed all the households that reported births in the past five years. Finally, we selected 37 households from each *limatbudin;* which amounts 8880 households in total from 240 *limatbudins*.

## Variables

Maternal mortality was the outcome measurement of the study. Explanatory variables were: educational level of mother, educational level of husband, road type used to reach the nearest health facility, distance to the nearest health centre, distance to the nearest hospital, occupation of household head, number of births given in past five years, family size, wealth index, hospital to population ratio, health centre to population ratio, doctor to population ratio and midwife to population ratio.

The geographic locations of the households, the nearest health centres and the nearest hospitals were mapped with a global positioning system (GPS) receiver by data collectors who visited all the sampled households during data collection. Traveling time by walking to the nearest hospital was assessed by the data collectors based on reports from the respondents.

Data on number of hospitals, health centres, doctors and midwives of the sampled and other districts of the region was obtained from Sidama National Regional Health Bureau, Human Resource Department (unpublished).

Wealth index was created using 15 household asset variables [18] broadly categorized in five groups: assets owned (radio, mobile phone and motorbike), livestock owned (cattle, horse or mule or donkey and sheep or goat), housing characterstics and utilities (flooring materials, roofing materials, number of rooms used for sleeping, source of drinking water, type of toilet facilities, access to electricity and use of kerosene lamp), cash crop grown and ownership of horse or mule used for transportation. Household utilities and asset variables used for

household wealth index creation are presented in S1 Table. Type of road to the nearest health facility was obtained from the report of participant interview.

## Definitions

*Maternal death*. A death of a woman while pregnant or within 42 days of termination of pregnancy, irrespective of the duration and site of the pregnancy, from any cause related to or aggravated by the pregnancy or its management, but not from accidental or incidental causes; International classification of diseases and related health problems (ICD-10) [24].

*Late maternal death*. A death of a woman from direct or indirect obstetric causes, more than 42 days but less than one year after termination of pregnancy [24].

*Comprehensive maternal death*. A grouping that combines both early and late maternal deaths (ICD-11) [25].

*Maternal mortality ratio (MMR)*. Is the number of maternal deaths during a given time period per 100,000 live births during the same time period.

*Verbal autopsy for maternal health*. A method of finding out the medical causes of death and ascertaining factors that may have contributed to the death in women who died outside of a medical facility. The VA consists of interviewing people who know about the events leading to the death such as family members, neighbours and traditional birth attendants [26].

## Data sources and measurement

The data was collected from households that reported pregnancy and pregnancy outcomes in the past five years. In a household which did not have maternal death, a mother was interviewed about her pregnancy experiences and household characteristics using interviewer administered questionnaire. When a mother was absent during the initial visit, the data collectors revisited the household the next day. The data was collected by diploma level teachers recruited from each kebele.

In a household where maternal death occurred, we interviewed a father or any adult knowledgeable about the death of a mother. The data was obtained through administering VA questions adapted from the WHO manual for maternal death [27]. Two public health officers who were familiar with the language and culture of study area independently conducted the VA interview. The VA interviewers determined the cause of death using pre-coded options of major causes of maternal deaths: bleeding (haemorrhage), fever (sepsis), convulsion (hypertension), prolonged or obstructed labour and including the option of other causes [24].

## Data quality control

The questionnaire was developed after reviewing similar studies. Initially, the questionnaire was prepared in English, translated into local language (*Sidaamu Afoo*) and then back translated to English by another individual. VA interview questions were adapted from the World Health Organization (WHO) VA guideline [27]. We used the WHO ICD-10 guideline for the ascertainment of causes of maternal deaths [24].

Inter-rater agreement between the two VA interviewers while ascertaining the cause of maternal deaths was assessed by kappa statistic. We used the Landis and Koch inter-rater reliability classification to interpret the kappa coefficient: < = 0.4: poor to fair; >0.4-< = 0.6: moderate agreement, >0.6–0.8: substantial agreement and >0.8-high agreement [28]. The computed Kappa statistics test result was Kappa = 0.75 (95% CI: (0.62–0.87) which indicates substantial agreement between the two VA interviewers.

Internal consistency of the variables used for wealth index creations was determined using Cronbach's Alpha reliability statistics which was found 0.54 and the sampling adequacy was assessed by Kaiser-Meyer-Olkin test with test result of 0.64.

The data collectors, the supervisors and VA interviewers were given training by the principal investigator. Key terms and concepts were translated into local terms during the training. The questionnaire was pretested in one district not included in the survey.

The supervisors followed the data collectors, checked consistency and completeness of the questionnaire on daily basis. The data was double entered and validated using EpiData version 3.1 software (EpiData Association 2000–2021, Denmark).

## Sample size estimation

Sample size estimation for the survey was determined based on the following assumptions: MMR of 412/100,000 LB, crude birth rate of 32 per 1000 population and average household size of 4.6 [3]. With the assumption of a MMR of 412 per 100,000 LB, we used design effect of 2 (as the study employed multistage cluster sampling method) and 0.14% precision level to obtain the number of LB needed for this study. The estimated sample was 15879 LB. We wanted to estimate maternal mortality within 0.14 percent point of the true value with 95% confidence.

From a population of 100,000 people and assumed crude birth rate of 32 per 1000 people, we would have (32/1000*100000) 3200 LB per year (16000 LB in 5 years). Hence, we expected to observe 66 maternal deaths over five years among 16000 LB with 95% confidence interval of MMR; 412 (324–524) per 100,000 LB [29].

We assumed that two LB would occur in one household over a five-year period [18] and hence 8000 households would be visited to get the 16000 LB. By considering 10% non-response, the final households estimated for the survey were 8800 households. We used Open-Epi software to calculate the sample size (Source Epidemiologic Statistics for Public Health version 3.01, www.OpenEpi.com) [29].

## Statistical analysis

We used Stata version 15 for data analysis (Stata Corp., LLC. College Station, Texas, USA). This study used data obtained through multistage cluster sampling design [30, 31]. To account for the sampling design, we employed complex survey data analysis method with sampling weight adjusted for non-response [30, 32]. The sampling weight was employed to correct for unequal probability of selection so that to produce meaningful estimates which correspond to the population of interest [33].

This study had four sampling units: district, *kebele*, *limatbudin* and household. In primary sampling unit, we applied similar sampling weight since the districts were selected with equal probability of selection. However, the *kebeles*, *limatbudin* and households were selected with different selection probability at their respective levels and hence we computed the sampling weight for each of them that differ according to their sampling probability.

We computed sampling weight adjusted for non-response by using three steps stated below [32]. We initially calculated the sampling weight for each sampling unit. The sampling weight was computed as the inverse of selection probability. Secondly, we adjusted for non-response for each sampling unit. Nonresponse was calculated as the inverse of response rate. Finally, we calculated sampling weight adjusted for non-response by multiplying the inverse of sampling probability (inverse of inclusion probability) with the inverse of response rate at each sampling unit [32].

We also estimated finite population correction (FPC) factor for each sampling unit to adjust for variance estimators as the survey data was sampled from finite population without replacement [34]. The FPC was calculated using the following formula where N is population and n is

sample:

$$\mathbf{FPC} = \sqrt{(\mathbf{N}-\mathbf{n})/(\mathbf{N}-\mathbf{1})}$$

Principal component analysis (PCA) was computed to create wealth index [35]. We categorized the wealth index using the first principal component with eigenvalue of 2.3 that explained 15.2% of the total variance.

We used geographic coordinates of households, the nearest health centres and hospitals to calculate distance between them. We calculated straight-line distances using proximity analysis "generate near table function" in ArcGIS 10.4.1 [36] and exported the data to Stata 15 for further analysis. Walking time to the nearest hospital according to the participants' report was also used.

We did descriptive statistics like mean, proportions and ratios. Chi-square test was computed to test the association between the outcome variable and potential explanatory variables. Complex sample logistic regression analysis was used to measure the effect of explanatory variables with maternal mortality. We carried out both weighted and non-weighted analysis, but reported only weighted analysis.

## Ethical approval

The ethical approval for this study was obtained from institutional review board of Hawassa University College of Medicine and Health Sciences (IRB/015/11) and Regional Ethical Committee of Western Norway (2018/2389/REK vest). Support letter to respective district (*woreda*) health offices was obtained from Sidama National Regional State Health Bureau (formerly known Sidama Zone Health Department). Letter of permission to respective *kebeles* was sought from each *woreda* health office.

Informed written (thumb print and signed) consent was obtained from the study participant before interview. Participant identifiers were anonymized during data entry and analysis to maintain confidentiality of the participants.

## Results

### Background characterstics of study districts

Table 1 summarizes the background characterstics of the six districts included in the study. Daela and Wondogenet districts did not have a hospital. Doctor to population ratio in Aroresa district was about 1 per 26000 while there were no doctors in Daela and Wondogenet districts. The midwife to population ratio was 1 per 6200 in Hawassa Zuriya district whereas 1 per 52000 in Aroresa district and 1 per 45900 in Daela district. Hawassa Zuriya is the nearest district, 21 kilometre distant from the regional capital, Hawassa, whereas Aroresa is the farthest district situated 181 kilometre away from Hawassa.

### Background characterstics of study population

Table 2 shows background characterstics of the study participants. In this study we interviewed 8755 participants out of 8880 households visited, with response rate of 98.6%. On average there were 5 persons (ranging 1–14 persons) per household. Concerning educational status, 2304 (24.3%) of the mothers and 1467 (15.7%) of the husbands had no formal education. Subsistence farming was the main occupation for 6332 (71.8%) head of the households. To access the nearest hospital, 7653 (89.6%) of the families needed more than an hour of walking. The nearest health centre for 8050 (93.3%) of households was found within 5km distance, while the nearest hospital for 6440 (77.7%) households was found within 10km distance.

**Table 1. Background characterstics of study districts, Sidama National Regional State, southern Ethiopia, 2020.**

| | Districts | | | | | |
|---|---|---|---|---|---|---|
| | Aleta- Chuko | Aleta- Wondo | Aroresa | Daela | Hawassa- Zuriya | Wondo- genet |
| Total population | 212,721 | 246,556 | 104, 006 | 45, 938 | 168, 186 | 171,291 |
| Estimated births (3.46%) | 7360 | 8530 | 3599 | 1589 | 5819 | 5927 |
| Primary hospital available (Number) | 1 | 1 | 1 | 0 | 1 | 0 |
| Health centre available (Number) | 7 | 7 | 5 | 1 | 4 | 5 |
| Primary hospital to population ratio* | 1:212721 | 1:246556 | 1:104006 | No hospital | 1:168186 | No hospital |
| Health centre to population ratio** | 1:30389 | 1:35222 | 1:20801 | 1:45938 | 1:42047 | 1:34258 |
| Additional primary hospital needed | 1 | 2 | 0 | 1 | 1 | 2 |
| Additional health centre needed | 2 | 3 | -1 | 1 | 3 | 2 |
| Number of midwives available | 24 | 26 | 2 | 1 | 27 | 8 |
| Number of doctors available | 10 | 13 | 4 | 0 | 14 | 0 |
| Midwife to population ratio*** | 1 per 8863 | 1 per 9483 | 1 per 52003 | 1:45938 | 1 per 6229 | 1 per 21411 |
| Doctor to population ratio**** | 1 per 21272 | 1 per 18966 | 1 per 26002 | No physician | 1 per 12013 | No physician |
| Additional midwives needed (number) | 19 | 23 | 19 | 8 | 7 | 26 |
| Additional doctors needed (number) | 11 | 12 | 6 | 5 | 3 | 17 |
| Average distance from regional capital, Hawassa (km) | 61km | 64km | 181km | 175km | 21km | 25km |

Note: Ideal population-to-facility ratios

* 1:100,000

** 1:25,000

*** 1:5000

**** 1:10000, km: kilometre

Source: Sidama National Regional State Health Bureau, Human Resource Department annual report, 2020, (unpublished).

## Place and assistance during delivery

Table 3 shows place and assistance of delivery. We identified a total of 10851 births: 10602 (LB) and 249 stillbirths. On average, there were 1.2 births in a household in the past five years, and 56.2% of the births took place at home. Traditional birth attendants (TBAs) assisted 18.3% of the births, 38.0% were assisted by family or neighbour and 43.2% were assisted by skilled health personnel.

## Characterstics of deceased mothers

Table 4 describes characterstics of deceased mothers. The mean age of deceased mothers was 29 years. Twenty two (47%) of the deaths occurred in the 25–29 age group. Twenty eight (55%) of deceased mother had no formal education, 38 (84%) were house wives, 40 (80%) were multiparous, 32 (67%) had pregnancy related complaints, 22 (50%) attended antenatal care (ANC) and from those who had ANC check-up, 5 (24%) attended four or more ANC visits. Twenty-four (59%) of deceased mothers gave birth at home from which 10 (21%) were assisted by TBA and 14 (38%) were assisted by family or neighbours.

## Maternal deaths

Table 5 shows causes, time and place of maternal deaths. We registered 10602 LB and 48 maternal deaths yielding the overall MMR of 419 (95% CI: 260–577) per 100,000 LB. In addition there were 7 late maternal deaths. Haemorrhage was the most common 21 (41%) direct cause of maternal deaths followed by eclampsia 10 (27%). Direct obstetric causes were responsible for 89% of the deaths while indirect obstetric causes accounted for 11%. Thirty (59%)

**Table 2. Background characteristics of study participants, Sidama National Regional State, southern Ethiopia, 2020.**

| Variable | Unweighted (Number, %) | | Weighted** % |
|---|---|---|---|
| Total visited households | 8755 (98.6) | | |
| Educational level of mother | | | |
| No formal education | 2304 (26.3) | | 24.3 |
| Lower primary school (Grade 1–4) | 2672 (30.5) | | 31 |
| Upper primary school (Grade 5–8) | 2926 (33.4) | | 34.3 |
| High school or above | 853 (9.7) | | 10.3 |
| Educational level of husband | | | |
| No formal education | 1467 (16.8) | | 15.7 |
| Lower primary school (Grade 1–4) | 1822 (20.8) | | 20.9 |
| Upper primary school (Grade 5–8) | 3782 (43.2) | | 43.8 |
| High school or above | 1684 (19.2) | | 19.6 |
| Main occupation of head of the household | | | |
| Subsistence farming | 6332 (72.3) | | 71.8 |
| Not farming* | 2423 (27.7) | | 28.2 |
| Wealth quantile of household | | | |
| Lowest | 3510 (40.1) | | 43.5 |
| Middle | 1749 (20.0) | 19 | |
| Highest | 3496 (39.9) | 34.5 | |
| Road facility to the nearest health facility | | | |
| Concrete | 741 (8.5) | 7.1 | |
| All weather road | 4618 (52.7) | 54.8 | |
| Dry weather road | 3153 (36.0) | 35.7 | |
| No motorized road: no (%) | 243 (2.8) | | 2.4 |
| Walking time to nearest hospital | | | |
| Less than one hour | 1102 (12.6) | 10.4 | |
| 1hour -2hour | 4566 (52.2) | 56.1 | |
| 2hour-4 hour | 2464 (28.1) | 27.1 | |
| Above 4 hour | 623 (7.1) | 6.4 | |
| Distance to the nearest HC | | | |
| < = 4.9Km | 8050 (91.9) | 93.3 | |
| > = 5Km | 705 (8.1) | 6.7 | |
| Distance to the nearest HO | | | |
| < = 9.9Km | 6440 (73.6) | 77.7 | |
| > = 10Km | 2315 (26.4) | 22.3 | |

Note

* Includes small scale trade and government employed

** Non-response adjusted sampling weight, HC: Health centre, HO: Hospital, Km: Kilometre

mothers had died during labour or within 24 hours after delivery. Twenty-five (47%) mothers died at home while 17 (38%) died at a health facility

## Variations of maternal mortality ratio

Table 6 shows the maternal mortality ratio by districts. Aroresa district had the highest MMR: 1142; 95% CI: 693–1591 per 100,000 LB. The three districts with the highest MMR in deceasing order were Aroresa, Daela and Aleta Wondo. The MMR in Aroresa district was 10 times

**Table 3. Place and assistance during delivery, Sidama National Regional State, southern Ethiopia, 2020.**

| Total births* (N = 10851) | Unweighted (Number, %) | Weighted **% |
|---|---|---|
| Birth place | | |
| Home | 6406 (59.0) | 56.2 |
| Health Post | 62 (0.6) | 0.5 |
| Health Centre | 3025 (27.9) | 30.0 |
| Hospital | 1318 (12.1) | 12.8 |
| Private Clinic or Hospital | 40 (0.4) | 0.5 |
| Attendant of Delivery | | |
| Traditional birth attendant | 1851 (17.0) | 18.3 |
| Health Extension Worker | 62 (0.6) | 0.5 |
| Family or Neighbour | 4555 (42.0) | 38.0 |
| Nurse or Doctor or Health Officer | 4383 (40.4) | 43.2 |

Note

* Includes live births and stillbirth

** Non-response adjusted sampling weight

higher to Hawassa Zuriya district and 4 times higher to Wondogenet district. Similarly the MMR in Daela district was 5 fold to Hawassa Zuriya district and more than double to Wondogenet district.

## Factors associated with maternal mortality

Table 7 shows results of complex sample logistic regression analysis of risk factors for maternal deaths. The risk of maternal death was higher among mothers without formal education than among those with formal education (AOR 4.4; 95% CI 1.7–11.0). Also, the risk of maternal death was higher in districts with low midwife-to-population ratio than those with high midwife-to-population ratio (AOR 2.9; 95% CI 1.0–8.9).

## Discussion

### Principal findings

In a population-based survey in the Sidama National Regional State, southern Ethiopia, we found an overall MMR of 419 per 100,000 LB with great variation by district. The most remote districts far from the central city, with poor infrastructure and inadequate skilled health personnel had the highest maternal mortality ratio compared to those districts found nearer to the central city, with good infrastructure and adequate skilled health personnel. Haemorrhage and eclampsia were the leading causes of death. Nearly half of maternal deaths occurred at home and about two fifths in health facilities. The risk of maternal death was high among mothers who had no formal education and in districts which had low midwife to population ratio.

### Strengths and weaknesses of the study

To the best of our knowledge, this is the first population based study describing maternal mortality estimate with district level variations in Sidama National Regional State, southern Ethiopia using large and representative sample. We used data collectors recruited from the study area that enhanced understanding and trustworthy communication with the study population. Each maternal death was independently reviewed by two public health officers using standard

**Table 4. Characteristics of deceased mothers, Sidama National Regional State, southern Ethiopia, 2020.**

| Variables | Unweighted Number (%) | Weighted* % |
|---|---|---|
| All deceased mothers | 48 (100) | |
| Age group (years) | | |
| 20–24 | 3 (6) | 7 |
| 25–29 | 22 (46) | 47 |
| 30–34 | 15 (31) | 30 |
| 35–39 | 6 (13) | 11 |
| 40–44 | 2 (4) | 4 |
| 45–49 | 0 (0) | 0 |
| Education level | | |
| No formal education (number, %) | 28 (58) | 55 |
| Formal education (number, %) | 20 (42) | 45 |
| Occupation | | |
| House wife | 38 (79) | 84 |
| Subsistence farming | 10 (21) | 16 |
| Parity | | |
| Nulliparous | 8 (17) | 20 |
| Multipara | 40 (83) | 80 |
| Complaints in the last three months | | |
| Had complaints | 32 (67) | 67 |
| Did not have complaints | 16 (33) | 33 |
| ANC attendance | | |
| Attended ANC | 22 (46) | 50 |
| Did not attend ANC | 26 (54) | 50 |
| Number of ANC (N = 22) | | |
| < = 3 | 17 (77) | 76 |
| > = 4 | 5 (23) | 24 |
| Place of delivery while died (N = 37) | | |
| Home | 24 (65%) | 59 |
| Health centre | 5 (13%) | 16 |
| Hospital | 8 (22%) | 25 |
| Assistance of delivery while died (N = 37) | | |
| TBA | 10 (27%) | 21 |
| Family | 9 (24%) | 23 |
| Neighbour | 5 (14%) | 15 |
| Nurse/Health officer/Doctor | 13 (35%) | 41 |

Note
* Non-response adjusted sampling weight

VA guidelines which improved precise assignment of cause of maternal deaths. To account for the multi-stage cluster sampling technique employed for the study, we used survey data analysis methods which improved precision of the estimates.

This study had some limitations. We studied maternal deaths that occurred in past five years, hence, recall bias and underreporting were important limitations of this study. However, to minimize recall bias, we used local calendar and events, that helped the respondents recognize the time of maternal death. Secondly we employed data collectors who were familiar with

**Table 5. Causes, time and place of maternal deaths, Sidama National Regional State, southern Ethiopia, 2020.**

| Causes of death | Unweighted Number, (%) | Weighted* % |
|---|---|---|
| All deaths | 48 (100) | |
| Direct obstetric causes | | |
| Haemorrhage | 21 (44) | 41 |
| Eclampsia | 10 (21) | 27 |
| Obstructed labour | 8 (17) | 14 |
| Retained placenta | 3 (6) | 6 |
| Sepsis | 1 (2) | 0.9 |
| Indirect obstetric Causes | | |
| Tuberculosis | 2 (4) | 5 |
| Malaria | 1 (2) | 3 |
| Diabetic Mellitus | 1 (2) | 2. |
| Others** | 1 (2) | 0.9 |
| Time of death (N = 48) | | |
| While pregnant | 7 (15) | 20 |
| During labour | 4 (8) | 6 |
| Within 24 hour after delivery | 26 (54) | 53 |
| Within 6 weeks after delivery | 11 (23) | 21 |
| Palace of death (N = 48) | | |
| At home | 25 (52) | 47 |
| On the way to health facility | 6 (13) | 15 |
| In health facility | 17 (35) | 38 |

Notes: * Non-response adjusted sampling weight

** Others include Leukaemia

the study setting and who took part in social events who supported the respondents to recall the death of mothers. In spite of our efforts to circumvent recall bias, there might be some maternal deaths not reported.

Though this study was done in one of the regional states of Ethiopia, we assume that the region could represents other regional states of the country in terms of health services and demographics. We also believe that the study was done using representative sample of the region as we followed probability sampling techniques in each sampling stage.

Misclassification could be another limitation for this study. Unlike medically certified deaths, our conclusion on causes of maternal deaths was based on lay family member report which is liable to misclassification. The following measures were taken to reduce misclassification. Firstly, we used two independent VA interviewers to ascertain cause of maternal deaths. Secondly, in case of lack of consensus between the two interviewers, we used third VA interviewer. Thirdly, when we did not get clear information from the first interviewee, we interviewed more than one family member.

We did not ask about deaths that occurred during early pregnancy due to abortion as we did not get ethical approval from the Regional Committee for Medical and Health Research Ethics (REK Western Norway) to include abortion in our study. Studies conducted in south-west Ethiopia estimated that maternal deaths ascribed to abortion accounted 8–10% of maternal deaths [37, 38]. There might be abortion related maternal deaths which were not reported in our study. Hence, we believe that the MMR was underestimated as we did not include abortion in our study.

We did not also ask about early pregnancy maternal deaths due to ectopic pregnancy since ascertaining ectopic pregnancy could be difficult in a rural setting. A study from Tigray

**Table 6. Maternal mortality ratio by district, Sidama National Regional State, southern Ethiopia, 2020.**

| Woreda | Livebirths | Maternal deaths | Unweighted | | Weighted* | |
|---|---|---|---|---|---|---|
| | | | MMR | 95% CI | MMR | 95% CI |
| Aleta Chuko | 2265 | 5 | 221 | 27–414 | 263 | 58–467 |
| Aleta Wondo | 2285 | 12 | 525 | 229–822 | 525 | 207–842 |
| Aroresa | 1694 | 21 | 1240 | 711–1768 | 1142 | 693–1591 |
| Daela | 716 | 4 | 559 | 11–1106 | 641 | 77–1358 |
| Hawassa Zuriya | 2237 | 2 | 89 | 35–213 | 114 | 24–251 |
| Wondogenet | 1405 | 4 | 285 | 5–564 | 258 | 8–508 |
| Over all Sidama | 10602 | 48 | 453 | 325–581 | 419 | 260–577 |

Note

*: Non-response adjusted sampling weight. Abbreviations; CI: Confidence interval, MMR: Maternal mortality ratio per 100000 live born

Region, northern Ethiopia showed that the prevalence of ectopic pregnancy was 0.52% of the total deliveries [39]. Though we assume that the prevalence of ectopic pregnancy is to be low, there might be ectopic pregnancy related maternal deaths which were not reported in our study.

All mothers in surveyed households were married women and we did not find single or women who were not in a marital union in our study. We believe that majority of pregnancies in rural community are a result of marriage. However, there might be maternal deaths in single or unmarried women which were not identified and reported by our study. A study from eastern Ethiopia reported that maternal deaths among never married women constituted 1 (2.4%) [40].

We lack some data on health system and other factors that might have contributed for variations of maternal deaths in the study districts. Due to resource limitation, we did not use software assisted VA algorisms and expert panel of obstetrician to ascertain the deaths. However, we provided adequate training for VA interviewers, pilot tested the questionnaire and the VA interviews were conducted by two independent interviewers.

Another limitation which we can mention is, in this study we found low birth rate than we had planned initially. We also noted that there were differences in birth rates across the districts in the region. This shows that the true birth rate in the region is lower than we had expected at first. A recent study conducted in Sidama National Region State is in agreement with our finding that found the fertility in the region has shown a falling trend [41].

In this study we were not able to find the number of maternal deaths we had anticipated initially. Our aim was to find 66 maternal deaths. However, we registered 48 maternal deaths. Our results also show that we have a wide 95% CI as we estimated MMR of 419 (95% CI: 260–577). A limitation of our study is thus reduced sample size. Though it is costly to attain precise estimate of MMR, since it needs a large number of maternal deaths, we could have obtained more precise estimate of MMR if our sample was larger than the current one.

## Magnitude and district level variations of maternal mortality

This study identified the overall MMR 419 per 100,000 LB. Our finding is in agreement with previous MMR estimates in Ethiopia [4, 18, 42]. However; its higher than the 2017 global average, 211 per 100,000 LB [4]. To achieve the SDG, countries must reduce their MMRs by at least 6% each year between 2016 and 2030; in Ethiopia between 2000 and 2017 the annual rate of MMR reduction was 5.5% [4].

We observed a very high MMR in Aroresa district. Similar finding has been reported from a study in Tigray region, northern Ethiopia [19]. Provincial differences in MMR has also been

**Table 7. Complex sample logistic regression analysis for factors associated with maternal mortality, Sidama National Regional State, southern Ethiopia, 2020.**

| Explanatory variables | Number of Women (8755) | Number of maternal deaths (48) | Crude OR | Logistic Regression | Complex sample * Logistic Regression |
|---|---|---|---|---|---|
| Mother education | | | COR (95% CI) | AOR (95% CI) | AOR (95% CI) |
| Had formal education | 6451 | 18 | 1 | 1 | 1 |
| No formal education | 2304 | 30 | 4.7 (2.6–8.5) | 3.1 (1.3–7.1) | 4.4 (1.7–11.0) |
| Husband education | | | | | |
| Had formal education | 7288 | 31 | 1 | 1 | 1 |
| No formal education | 1467 | 17 | 2.7 (1.5–5.0) | 1.1 (0.5–2.3) | 0.9 (0.4–2.0) |
| Main occupation of household head | | | | | |
| Not farming** | 2423 | 8 | 1 | 1 | 1 |
| Farming | 6332 | 40 | 1.9 (0.9–4.1) | 1.7 (0.7–4.2) | 1.9 (0.7–5.0) |
| Family size | | | | | |
| < = 4 | 3958 | 15 | 1 | 1 | 1 |
| > = 5 | 4797 | 33 | 1.8 (1.0–3.4) | 1.4 (0.7–3.1) | 1.6 (0.7–3.7) |
| Place of birth (8744)*** | | | | | |
| Health institution | 4894 | 13 | 1 | 1 | 1 |
| Home | 3850 | 24 | 0.7 (0.3–1.3) | 1.5 (0.7–3.2) | 1.8 (0.8–4.2) |
| Wealth index household | | | | | |
| Rich | 5245 | 27 | 1 | 1 | 1 |
| Poor | 3510 | 21 | 1.2 (0.6–2.1) | 1.5 (0.7–2.9) | 1.5 (0.7–3.3) |
| Distance to the nearest HC | | | | | |
| < = 4.9Km | 8050 | 41 | 1 | 1 | 1 |
| > = 5Km | 705 | 7 | 1.9 (0.9–4.4) | 1.3 (0.5–3.7) | 1.4 (0.3–6.0) |
| Distance to the nearest HO | | | | | |
| < = 9.9Km | 6440 | 31 | 1 | 1 | 1 |
| > = 10Km | 2315 | 17 | 1.5 (0.8–2.8) | 1.1 (0.4–2.6) | 1.1 (0.5–2.5) |
| Hospital to population ratio | | | | | |
| Standard or above (> = 1) | 1305 | 21 | 1 | 1 | 1 |
| Below standard (<1) | 7450 | 27 | 0.6 (0.4–1.1) | 0.3 (0.1–1.0) | 0.3 (0.1–1.2) |
| Doctor to population ratio | | | | | |
| Above mean (>0.4) | 3950 | 14 | 1 | 1 | 1 |
| Mean and below (< = 0.4) | 4805 | 34 | 2.0 (1.1–3.7) | 1.0 (0.3–3.4) | 1.1 (0.3–3.4) |
| Midwife to population ratio | | | | | |
| Above mean (>0.5) | 3941 | 7 | 1 | 1 | 1 |
| Mean and below (< = 0.5) | 4814 | 41 | 4.8 (2.2–10.8) | 3.4 (1.0 11.6) | 2.9 (1.0–8.9) |

Note

* Non-response adjusted sampling weight

**Includes small scale trade and government employed

*** 11 mothers had died before giving birth, AOR: Adjusted odds ratio, CI: Confidence interval. COR: Crude odds ratio, HC: Health Centre, HO: Hospital, OR: Odds ratio

reported from other African country [43]. Low utilization rate of maternal health services might have contributed for the high number of maternal deaths in the district. For instance, a study by Limaso et al. documented that in 2018 the coverage of institutional delivery for Aroresa district as reported from the district health office was 38% [44]. Aroresa district is the most remote district in the region situated 181 Km distant from the regional capital [44]; most of

the kebeles in this district have difficult topography and poor road conditions, where health facilities were hard to reach. Weak referral system and lack of emergency transportation might have contributed for the high MMR in the district. Aroresa district had the lowest midwife-to-population ratio among the 6 districts included in the study. Remoteness, distance of health facilities from households and lack of adequate and skilled health personnel are known risk factors for maternal death [45].

## Time, cause and place of maternal mortality

In our study we found that about 60% of maternal deaths occurred during labour or within 24 hours postpartum. This finding is in agreement with a study conducted in eastern part of Ethiopia where 55.6% of maternal deaths were reported to occur within the first day [40]. The time around labour and the first 24 hour postpartum is a critical period that a mother should get emergency obstetric care at health facility.

Haemorrhage and hypertensive disorder of pregnancy accounted for majority of the deaths. This is in agreement with the findings that have been reported from Ethiopia and elsewhere [40, 46–48]. Recent studies have indicated that the prevalence of post-partum haemorrhage in Ethiopia ranges 7.6% to 16.6% [49, 50] and hypertensive disorders of pregnancy (pre-eclampsia/eclampsia) 2–3% to 12.4% [51–53].

We observed that around 50% of maternal deaths occurred at home. This is in agreement with the study reported from eastern part of Ethiopia where 56% maternal deaths were found to occur at home [40]. The high proportion of maternal deaths that occurred at home could be associated with the low coverage of skilled birth attendant which we observed in the study area. Home deaths might be reflection of poor access to emergency obstetric care.

In this study we found significant number of maternal deaths occurred in health facilities. A study from eastern part of Ethiopia found similar finding [40]. A study from Indonesia documented that poor quality of care at health facility was associated with high chance of maternal deaths [54]. Poor and inadequate emergency obstetric care at health facilities might have contributed for the deaths occurred at health facilities [55–57].

## Skilled delivery

This study found that more than half of the births took place at home assisted by either TBA, family or neighbours. Studies documented that community trust on TBA, lack of transportation and poor quality of maternal health services were associated with low utilization of maternal health services [58]. It has been reported skilled assistance at delivery associated with less maternal deaths [59, 60].

## Independent predictors of maternal mortality

Mothers with no formal education had increased risk of maternal death. A study conducted in eastern part of Ethiopian showed that around 84% deceased mothers were illiterate [40]. The association of low education level with severe maternal outcomes has been documented [61]. A multi-country study showed mothers with no education had 2.7 times higher risk of mortality than mothers with high education [62]. Educated women better utilize maternal health services [63, 64] and recognize pregnancy complications, prepare for births and obstetric emergencies [65].

In this study, we observed that there was increased risk of maternal mortality in districts which had inadequate number of midwives compared with districts which had adequate number of midwives. It has been documented that availability of skilled midwives at health facilities increases the uptake of institutional deliveries and other maternal health services [66] and

consequently reduce the maternal mortality [14, 67]. In contrary to this; lack of skilled personnel for maternal health services increases the risk of maternal deaths. An Indonesian study documented that lack of adequate number of doctors working at community health centre and village was associated with high maternal mortality [45].

Some studies indicated low wealth status is associated with increased risk of maternal deaths [68]. However, in this study we did not see association of wealth status with maternal death. Similar finding has been reported by a study from southwest Ethiopia [18]. This study also found that place of birth was not associated with maternal death. Lack of association of wealth status and place of birth with maternal deaths could be explained by maternal death is rare event in terms of absolute number.

## Policy and clinical implications

This study from Sidama National Regional State highlights that Ethiopia needs more regional studies to address high maternal mortality rates in the country. The high MMR with significant district level variations in this study indicate there is a need to amplify the efforts to decease maternal deaths, identify risk factors and institute interventions tailored to areas with high maternal mortality in the region. Similar to other studies, haemorrhage was the leading cause of maternal deaths in the region. This highlights the importance of improving the skill and practice of health workers in active management of third stage of labour. The Sidama National Regional State Health Bureau as well as Ministry of Health should also consider inclusion of misoprostol provision to postpartum bleeding with the health extension packages for women who give birth without a skilled provider [69, 70]. Comprehensive emergency obstetric care including blood transfusion services should be available within the reach of community.

The high number of deaths attributable to hypertensive disorders of pregnancy in this rural community indicates the necessity of improving screening and detection of preeclampsia at community level using the health extension workers [71]. Referral system and management of preeclampsia/eclampsia has to be strengthened in the region.

In our study many maternal deaths occurred at home without skilled birth attendants which shows the need for strengthening access to emergency obstetric care. Evidences from rural settings of Ethiopia showed that interventions focused on strengthening emergency obstetric care improved the uptake of maternal health services and decreased the maternal mortality [15]. During ANC visits, pregnant women should be counselled and encouraged for skilled birth attendant. Avoiding barriers to skilled delivery and integrating the services of TBA with formal health system may increase use of skilled birth attendants [72].

The occurrence of significant number of maternal deaths at health facility signals the importance of improving emergency obstetric care in health facilities [57, 73]. The association of inadequate number of midwives with maternal mortality indicates the Ministry of Health and the Sidama National Regional State Health Bureau should train and deploy adequate number of midwives so that quality maternal health services are provided and consequently maternal deaths averted. In addition, the assignment and distribution of midwives and doctors should be fair so that the gap between the central and remote districts in distribution of skilled health personnel be minimized. Increased risk of maternal deaths among mothers who did not have formal education indicates there is a need to improve educational status of female in the region.

## Conclusion

This study found high MMR in rural areas of Sidama National Regional State with high variations in the districts. The study highlights that Ethiopia needs regional studies to understand

magnitude of maternal mortality and local variations in order to reduce the high maternal deaths. The quality of emergency obstetric care including lifesaving interventions have to be improved in the region. Sidama National Regional State Health Bureau should design maternal health interventions targeting local variations and areas with high mortality rates. The shortage of midwives should be alleviated to improve provision of skilled maternal health services and consequently save the life of mothers.

## Supporting information

**S1 Table. Variables included in principal component analysis for household wealth index creation, Sidama Regional State, southern Ethiopia, 2020.**
(XLSX)

## Acknowledgments

We would like to thank participants of the study for their time and information. Our special thanks go to Sidama National Regional State Health Bureau, respective district health offices and *kebele* administrations in Ethiopia for their support and permission to conduct the study. We are grateful to data clerks for their patience and time while dealing with big data without compromising its quality.

## Author Contributions

**Conceptualization:** Aschenaki Zerihun Kea, Bernt Lindtjorn, Sven Gudmund Hinderaker.

**Data curation:** Aschenaki Zerihun Kea.

**Formal analysis:** Aschenaki Zerihun Kea.

**Investigation:** Aschenaki Zerihun Kea, Bernt Lindtjorn, Sven Gudmund Hinderaker.

**Methodology:** Aschenaki Zerihun Kea, Bernt Lindtjorn, Sven Gudmund Hinderaker.

**Project administration:** Aschenaki Zerihun Kea.

**Supervision:** Bernt Lindtjorn, Achamyelesh Gebretsadik, Sven Gudmund Hinderaker.

**Writing – original draft:** Aschenaki Zerihun Kea.

**Writing – review & editing:** Aschenaki Zerihun Kea, Bernt Lindtjorn, Achamyelesh Gebretsadik, Sven Gudmund Hinderaker.

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
