## [Decision Letter · Decision Letter 0]

28 Nov 2022

PONE-D-22-19620Variation in maternal mortality in Sidama Regional State, southern Ethiopia: A population based cross sectional household surveyPLOS ONE

Dear Dr. Aschenaki Zerihun Kea,

Thank you for submitting your manuscript to PLOS ONE. After careful consideration, we feel that it has merit but does not fully meet PLOS ONE’s publication criteria as it currently stands. Therefore, we invite you to submit a revised version of the manuscript that addresses the points raised during the review process.

We look forward to receiving your revised manuscript.

Kind regards,

Sebsibe Tadesse, PhD

Academic Editor

PLOS ONE

https://journals.plos.org/plosone/s/fileid=ba62/PLOSOne_formatting_sample_title_authors_affiliations.pdf.

a. You may seek permission from the original copyright holder of Figure1 to publish the content specifically under the CC BY 4.0 license.   

Natural Earth (public domain): http://www.naturalearthdata.com/.

Reviewers' comments:

Reviewer's Responses to Questions

**Comments to the Author**

1. Is the manuscript technically sound, and do the data support the conclusions?

Reviewer #1: Partly

Reviewer #2: Yes

2. Has the statistical analysis been performed appropriately and rigorously? 

Reviewer #1: Yes

Reviewer #2: Yes

3. Have the authors made all data underlying the findings in their manuscript fully available?

Reviewer #1: No

Reviewer #2: Yes

4. Is the manuscript presented in an intelligible fashion and written in standard English?

Reviewer #1: No

Reviewer #2: Yes

5. Review Comments to the Author

Reviewer #1: This is an interesting study on a highly relevant topic. However, I am afraid that it does not really yield a valid MMR for Sidama Regional State, as only 6 out of 30 districts were selected, with large variations in district-specific MMRs. Further, the data quality is somewhat questionable, as the authors discuss. Nevertheless, I think the study has some value as the AORs are still informative, and it might serve as an example for carrying out similar studies in other regions of Ethiopia. More detailed comments are listed below.

- l. 71-72: The two short sentences seem somewhat superfluous. It might be more interesting what is likely to explain the reduction in MMR and whether this trend is restricted to specific regions in Ethiopia (if known).

- l. 80: It might be helpful to the reader if some information can be added why exactly this region was chosen for MMR assessment.

- l. 111-118: This information should rather be put into the Results section.

- Table 1: Does "estimated births" mean that the exact number of births was not known? If so, how could it be justified to assume a uniform birth rate of 3.46% (per year?) for each district?

- l. 129-130: I am afraid that choosing 6 out of 30 districts might not be sufficient to yield representative results for the whole region, even if the selection was done randomly. How do these 6 districts compare to the other 24 districts with respect to the major variables from table 1?

- l. 213-225: This paragraph needs some clarification: What is meant by a design effect of 2 and a precision level of 14%, and why were exactly these values chosen?

- I do not understand why a PCA was needed to calculate the wealth index - why was it not just calculated as a summary measure of the variables listed in table S1?

- Table 2: Were all mothers who were investigated in this survey indeed married, i.e. were pregnancies in single or unmarried mothers not taken into account? If so, might this potentially have biased the MMR estimates?

- Table 3: It seems that the real number of births per household was heavily overestimated in the sample size calculation. What was the reason for this, and how did this affect the results?

- Were there no maternal deaths which were associated with stillbirths?

- Some proofreading might be necessary, as there are some typos throughout the manuscript (that's why I answered "No" to question 4).

- In the spirit of Open and Reproducible Science, the analysis code should be made available in an online repository together with a data dictionary, and the respective URL should be mentioned in the Methods section.

Reviewer #2: Mortality In Sidama State, Ethiopia

General comments

This is an interesting article and the methods are sound. It deserves to be published after dealing with minor corrections.

The only main flaw is that the authors state “We did not ask about deaths that occurred during early pregnancy due to ectopic pregnancy or abortion.” why not? This is an important omission.

In the discussion, the authors need to explain why there is such a massive variation in availability of doctors and midwives. How can there be districts in the same region with 14 doctors and 27 midwives but another has 0 doctor and only 1 midwife, for a whole district?? Surely this in itself should be enough to motivate action and redistribution of resources, without even needing to do any further study on maternal deaths!

Corrections needed:

There are several mistakes in the English which need to be corrected. I don’t have time to list them all, but some examples are listed below. I would recommend review by a native English speaker.

Abstract

• Aroresa districts or Aroresa district? Is it 1 or more districts?

• with 24 hours after delivery or within 24 hours after delivery?

• “Additional midwives has to be trained” – change to “Additional midwives have to be trained”

Intro

• “Improving maternal health and consequently decrease maternal mortality is one of the important agenda for the government”

• Change to “Improving maternal health and consequently decreasing maternal mortality is important for the government” OR “Important priorities on the government’s agenda include Improving maternal health and consequently decreasing maternal mortality”

• “All women who experienced pregnancy and birth outcomes in the past five year in Sidama

• Regional State were the source population. Women residing in a sampled households… ”

• Should be ““All women who experienced pregnancy and birth outcomes in the past five years in Sidama Regional State were the source population. Women residing in sampled households… ”

• line 327 “with 24 hours after delivery” should be “within 24 hours after delivery”

• line 369 “two fifth” should be “two fifths”.

Results – no need to repeat in text what is already shown in tables. Just summarise the main findings.

“Aroresa district is the most remote district in the region situated 181 Km distant from the regional capital, 181 km [36];” – don’t need to repeat 181km twice.

“An Indonesian study documented that mothers who received inadequate care at health facility were more prone to death than those survived from severe obstetric complications [47].” - this doesn’t make sense. Please rephrase. Of course they would be more prone to death than those who survived!!!

“This study found that more than half of the births took place at home assisted by either TBA, family or neighbours”. Why was this factor not included in your logistic regression analysis?

6. PLOS authors have the option to publish the peer review history of their article (what does this mean?). If published, this will include your full peer review and any attached files.

Reviewer #1: No

Reviewer #2: **Yes: **Dr Merlin Willcox

---

## [Author Response · Author response to Decision Letter 0]

6 Jan 2023

Date: 06-Jan-23

 PONE-D-22-19620

Variation in maternal mortality in Sidama Regional State, southern Ethiopia: A population based cross sectional household survey

PLOS ONE 

Point by point response to reviewers’ and editor comments and suggestions 

The authors would like to thank the reviewers and editor for their time, suggestions and valuable comments. We have carefully addressed all the comments and suggestions. The corresponding changes made in the revised manuscript are summarized in our response below.

Answer to Reviewer #1 comments 

Comment: L71-72: The two short sentences seem somewhat superfluous. It might be more interesting what is likely to explain the reduction in MMR and whether this trend is restricted to specific regions in Ethiopia (if known).

 Answer: We would like to thank the reviewer for pointing out this issue. We added a sentence describing key measures undertaken by the government “improving access to universal health coverage and emergency obstetric care that played key role in reduction of maternal mortality in the country in general” (Lines 73-76, Ref [17]). 

Studies indicating the trends in MMR reduction in specific regions of the country are scarce. An implementation study from south-west Ethiopia demonstrated that the MMR declined by 64% during the intervention period from 477 to 219 deaths per 100,000 LB. (Lines 86-88, Ref [15]). 

Comment: L. 80: It might be helpful to the reader if some information can be added why exactly this region was chosen for MMR assessment. 

 Answer: We agree with the reviewer’s comment. We describe our aims as follow why this study was conducted in Sidama Region. To improve maternal health services and consequently reduce the maternal mortality in the region there is a need to have data or information at regional and district level. However, the country’s maternal mortality data mostly comes from national level study which does not show local problem in detail. We also wanted to know if there are variations in maternal mortality in different geographical areas of the region. 

As Ethiopia is big country constituted by different regional states and districts, the lesson from Sidama National Regional States will help to carryout similar studies in other areas that gives useful information for local planning and decision making and support the country’s effort to attain the Sustainable Development Goal. 

In addition, we added the following sentence why we chose Sidama Region for maternal mortality assessment “As there is no previous population-based study describing maternal mortality estimates and district-level variations in Sidama National Regional State, and as the principal investigator (AZK) is affiliated with Hawassa University, which is located in Sidama National Regional State, it was natural to conduct such a comprehensive study on this population” (Lines 89-92). 

Though this study was done in one of the regional states of Ethiopia, we assume that the region could represents other regional states of the country in terms of health services and demographics. We also believe that the study was done using representative sample of the region by employing probability sampling techniques in each sampling stage. (Lines 431-434). 

Comment: L.111-118: This information should rather be put into the Results section.

 Answer: We agree with the reviewer’s comment. Hence, we moved the information (lines 294-301) and accompanying Table 1 (lines 311-312) from the method section to the result section. 

Comment: Table 1: Does "estimated births" mean that the exact number of births was not known? If so, how could it be justified to assume a uniform birth rate of 3.46% (per year?) for each district?

 Answer: We appreciate the comment forwarded by the reviewer. When we started our study, we initially collected background information of the study districts (Table 1) and other districts in the region. Concerning the number of births per district, we could not get reported babies born but we used pooled birth rates from Sidama National Regional State Health Bureau, and we cited the source of information as foot note in Table 1. The Regional Health Bureau uses primary data from the Central Statistical Agency of Ethiopia by adapting to its local context. 

The birth rate we found after the study was lower than what we obtained from the Sidama National Regional State Health Bureau, and also, we noted that there were differences across the districts. This shows that the true birth rate in the region is lower than we had expected (lines 464-467). A recent study conducted in Sidama Region agrees with our finding that found the fertility in the region has shown a falling trend (Lines 467-468, Ref [42]). 

 Comment: L. 129-130: I am afraid that choosing 6 out of 30 districts might not be sufficient to yield representative results for the whole region, even if the selection was done randomly. How do these 6 districts compare to the other 24 districts with respect to the major variables from table 1?

 Answer: We acknowledge the reviewer’s comment, and we admit that our sample size could have been larger. Below we describe the sampling procedures we followed during the selection of our sample.

We followed probability sampling techniques at each sampling stage. Every unit in the population had a known chance of being selected in the sample. At first stage, we listed all the 30 woredas (districts) and selected 6 (20%) of the districts by simple random sampling. The 6 districts had a total of 104 kebeles (the smallest administrative structure with average population of 5000) ranging from 5 kebeles in Daela district to 25 Kebeles in Aleta Wondo district. 

At the second stage, we listed all the kebeles in 6 sampled districts and selected 40 kebeles proportional to the number of kebeles in the districts. 

At third stage, we listed all the Limatbudin (administrative unit under the kebele) in 40 kebeles. A limatbudin consists of 40-50 neighboring households. The 40 Kebeles had a total of 961 Limatbudins and we selected 6 Limatbudins from each kebele (240 limatbudins in total).

 At fourth stage, in each Limatbudin, we listed all the households that reported pregnancy and birth outcomes in the past five years, and finally selected 37 households from each Limatbudin using simple random sampling method (8880 households in total from 240 Limatbudins).

Probability sampling technique is the gold standard method recommended to observe reliable findings (precision) (lines 141-143, Ref-[23]). To make inferences about the population from the observation made from our sample, we applied this standard technique (probability sampling technique) at each sampling stage of this study. 

Our aim was to find 66 maternal deaths with MMR; 412 (95% CI: 324-524) per 100,000 LB (lines 238-240). In our study, we registered a total of 10602 live births and 48 maternal deaths. We found lower maternal deaths than we anticipated. Our results show that we have a wide 95% CI as we estimated MMR of 419 (95% CI: 260-577). This could have been improved if we had studied larger sample. A limitation of our study is thus this reduced sample size, as we have written in discussion section (lines 469-474). 

Comment: L. 213-225: This paragraph needs some clarification: What is meant by a design effect of 2 and a precision level of 14%, and why were exactly these values chosen?

 Answer: We appreciate the reviewer’s comment. We would like to inform that 14% precision was written by mistake, and we corrected to 0.14% of precision. We rephrased the paragraph as follow: when we calculated the sample size for this study, the following assumptions were considered: maternal mortality ratio (MMR) of 412 per 100, 000 live births (LB), design effect of 2 (as the study followed multistage cluster sampling technique) and 0.14% precision. We wanted to estimate maternal mortality within 0.14 percent point of the true value with 95% confidence (lines 232-236). 

The design effect is a correction factor that was used to adjust required sample size for multistage cluster sampling. The required sample size was estimated assuming a random sample, and then multiplied by the design effect. This accounts for the loss of information inherent in the clustered design.

Comment: I do not understand why a PCA was needed to calculate the wealth index - why was it not just calculated as a summary measure of the variables listed in table S1?

 Answer: The wealth index commonly used in household surveys is a composite index composed of key asset ownership variables used as a proxy indicator of household level wealth. The variables need data reduction analysis commonly done by exploratory analysis or Principal Component Analysis (PCA). The PCA is recommended by scholars for household surveys and commonly used in Demographic and Health Surveys (we cited the reference (line 270, Ref-[35]).

Comment: Table 2: Were all mothers who were investigated in this survey indeed married, i.e. were pregnancies in single or unmarried mothers not taken into account? If so, might this potentially have biased the MMR estimates?

 Answer: We agree with the comment forwarded by the reviewer, and we have discussed this as a limitation of the study. All mothers in surveyed households were married and we did not find single or women who were not in a marital union in our study. We believe that majority of pregnancies in rural community are a result of marriage. However, there might be maternal deaths in single or unmarried women which were not identified and reported by our study (lines 454-457). A study from eastern Ethiopia reported that maternal deaths among never married women constituted 1 (2.4%) (Lines 457-458, Ref [40]. 

Comment: - Table 3: It seems that the real number of births per household was heavily overestimated in the sample size calculation. What was the reason for this, and how did this affect the results?

 Answer: Thanks for mentioning this and we admit the comment. We used a previous study to calculate the sample size (estimate number of births per household) for our study (line 241, Ref [18]). This had led us to underestimate our sample and consequently affected the MMR estimation. 

Our initial plan was to find 2 births per household (based on the estimation of the study we stated above) in the past five years before the study. However, the number of births we found per household in our study was lower than the findings of the study we used for sample size calculation. A recent study in Sidama Region indicated that the fertility in the region has shown a falling trend (Lines 467-468, Ref [41]). We have mentioned this as a limitation under discussion section (lines 464-467).

 Comment: Were there no maternal deaths which were associated with stillbirths?

 Answer: Yes, there were maternal deaths associated with stillbirths. Mothers who had pregnancy and birth outcomes (live births, still births and neonatal deaths) in the past five years before the survey were the study population and included in the estimation of MMR.

Comment: Some proofreading might be necessary, as there are some typos throughout the manuscript (that's why I answered "No" to question 4).

 Answer: We agree with this comment, and we did proofreading and corrected some typos throughout the manuscript 

Answer to Reviewer #2 comments 

Comment: The only main flaw is that the authors state “We did not ask about deaths that occurred during early pregnancy due to ectopic pregnancy or abortion.” why not? This is an important omission.

 Answer: Thanks for commenting on this and we accept the comment. Yes indeed, we didn’t ask about deaths that occurred during early pregnancy due to abortion. We mentioned this as limitation of the study in discussion section as stated below: “We did not ask about deaths that occurred during early pregnancy due to abortion as we did not get ethical approval from the Regional Committee for Medical and Health Research Ethics (REK Western Norway) to include abortion in our study (lines 442-444). Studies conducted in south-west Ethiopia estimated that maternal deaths ascribed to abortion accounted 8-10% of maternal deaths (lines 444-446, Ref [37.38]). There might be abortion related maternal deaths which were not reported in our study. Hence, we believe that the MMR was underestimated as we did not include abortion in our study (lines 446-448). 

We did not also ask about early pregnancy maternal deaths due to ectopic pregnancy since ascertaining ectopic pregnancy could be difficult in a rural setting (lines 449-450). “A study from Tigray region, northern Ethiopia have showed the prevalence of ectopic pregnancy was 0.52% of the total deliveries (lines 450-452, Ref [39]). Though we assume that the prevalence of ectopic pregnancy is to be low, there might be ectopic pregnancy related maternal deaths which were not reported in our study (lines 452-453). 

Comment: In the discussion, the authors need to explain why there is such a massive variation in availability of doctors and midwives. How can there be districts in the same region with 14 doctors and 27 midwives but another has 0 doctor and only 1 midwife, for a whole district??

 Answer: Thanks for pointing out this issue. As you can appreciate from Table 1, the availability of doctors and midwives correlated with the existing type and number of health facilities in a district. For instance, if there is no hospital in a district, doctors may not be assigned in that district. 

Another reason could be the Regional Health Bureau formerly known as the Zonal Health Department did not assign staff to the remote areas and staff wanted to work in more central places.

 On top of recommending to train and deploy additional midwives, we also recommended that there should be fair distribution of the health personnel (midwives and doctors) within the region to minimize the gap between the central and remote districts in terms of distribution of health professionals (lines 575-577). 

Comment: Corrections needed: There are several mistakes in the English which need to be corrected. I don’t have time to list them all, but some examples are listed below. I would recommend review by a native English speaker.

 Answer: Thanks for this comment. As we describe below, we incorporated (revised) the suggested grammatical errors and typos in respective comments. We also proofread the manuscript and corrected similar mistakes. 

Comment: Aroresa districts or Aroresa district? Is it 1 or more districts?

 Answer: Thanks, we have made correction as:”Aroresa district” (L- 31)

Comment: with 24 hours after delivery or within 24 hours after delivery?

 Answer: Thanks, we have corrected with “within 24 hours after delivery” (L- 34)

Comment: Additional midwives has to be trained” – change to “Additional midwives have to be trained”

 Answer: we have changed to “Additional midwives have to be trained” (L- 43)

Comment: “Improving maternal health and consequently decrease maternal mortality is one of the important agenda for the government”

 Answer: We revised the sentence as per the suggestion of the reviewer added the following “Important priorities on the government’s agenda include Improving maternal health and consequently decreasing maternal mortality” (L- 48-50)

Comment: “All women who experienced pregnancy and birth outcomes in the past five year in Sidama Regional State were the source population. Women residing in a sampled households… ”

 Answer: We made a revision as follow “All women who experienced pregnancy and birth outcomes in the past five years in Sidama Regional State were the source population. Women residing in sampled households and who had pregnancy and birth outcomes (live births, stillbirths and neonatal deaths) in the past five years preceding the survey were the study population” (L- 136-139)

Comment: line 327 “with 24 hours after delivery” should be “within 24 hours after delivery”

 Answer: We accepted and changed to “within 24 hours after delivery” (L- 369)

Comment: line 369 “two fifth” should be “two fifths”.

 Answer: We acknowledge the correction and changed accordingly to “two fifths”. (L- 412)

Comment: Results – no need to repeat in text what is already shown in tables. Just summarize the main findings. “Aroresa district is the most remote district in the region situated 181 Km distant from the regional capital, 181 km [36];” – don’t need to repeat 181km twice.

 Answer: Thanks for pointing out this and we have deleted repetition (L- 487)

Comment: An Indonesian study documented that mothers who received inadequate care at health facility were more prone to death than those survived from severe obstetric complications [47].” - This doesn’t make sense. Please rephrase. Of course they would be more prone to death than those who survived!!!

 Answer: We accept the comment and rephrased the sentence like this “A study from Indonesia documented that poor quality of care at health facility was associated with high chance of maternal deaths (L- 512-514, Ref-[54])

Comment: “This study found that more than half of the births took place at home assisted by either TBA, family or neighbors”. Why was this factor not included in your logistic regression analysis?

 Answer: We acknowledge the comment. Initially, we did not include place of birth into to logistic regression analysis as it was not significantly associated with maternal death in bivariate analysis. 

By considering the reviewer’s comment and the importance of the variable, we included it into logistic regression analysis (Table 7 page 25). However, the variable was not associated with maternal death. This could be explained by that maternal death is rare events in terms of absolute number.

However, in descriptive analysis (considering deceased mothers only) we described the proportion of maternal deaths from total deaths (home delivery versus institutional delivery). Among the deceased mothers, around 60% of them were mothers who had given birth at home (L- 345-346).

---

## [Editor Report · Decision Letter 1]

25 Jan 2023

Variation in maternal mortality in Sidama National Regional State, southern Ethiopia: A population based cross sectional household survey

PONE-D-22-19620R1

Dear Dr. Aschenaki Zerihun Kea,

We’re pleased to inform you that your manuscript has been judged scientifically suitable for publication and will be formally accepted for publication once it meets all outstanding technical requirements.

Kind regards,

Sebsibe Tadesse, PhD

Academic Editor

PLOS ONE

---

## [Editor Report · Acceptance letter]

27 Feb 2023

PONE-D-22-19620R1 

Variation in maternal mortality in Sidama National Regional State, southern Ethiopia: A population based cross sectional household survey 

Dear Dr. Kea:

I'm pleased to inform you that your manuscript has been deemed suitable for publication in PLOS ONE. Congratulations! Your manuscript is now with our production department. 

Kind regards, 

on behalf of

Dr. Sebsibe Tadesse 

Academic Editor

PLOS ONE